# Institutional delivery and its determinants among reproductive-age women in Mozambique: A geographic weighted regression

Alebachew Ferede Zegeye[1]*, Habtu Kifle Negash[2], Alemu Teshale[3], Mequanint Kassa[2], Woretaw Endalew[2], Belete Assefa[4], Araya Mesfin Nigatu[5], Solomon Gedlu Nigatu[5]

1 Department of Medical Nursing, School of Nursing, College of Medicine and Health Sciences, University of Gondar, Gondar, Ethiopia, 2 Department of Epidemiology, Institute of Public Health, College of Medicine and Health Sciences, University of Gondar, Gondar, Ethiopia, 3 Department of Obstetrics and Gynecology, U-PRS Fellow, School of Medicine, College of Medicine and Health Sciences, University of Gondar, Gondar, Ethiopia, 4 Department of Internal Medicine, School of Medicine, College of Medicine and Health Sciences, University of Gondar, Gondar, Ethiopia, 5 Department of Epidemiology and Biostatistics, Institute of Public Health, College of Medicine and Health Sciences, University of Gondar, Gondar, Ethiopia

* alexferede24@gmail.com

## Abstract

Institutional delivery in Mozambique remains a critical public health issue. Despite efforts to improve maternal healthcare, many women still give birth at home. Moreover, the spatial pattern and spatial variables linked to institutional delivery in regions of Mozambique have not yet been discovered. Thus, this study aimed to explore the geographical variation of institutional delivery and its determinants among reproductive-age women living in Mozambique using geographically weighted regression analysis. The most recent Demographic and Health Survey data from Mozambique were used for secondary data analysis. In this study, a sample of 5,983 reproductive-age women in Mozambique was used. Data cleaning and preparation were conducted using STATA version 17 and Microsoft Excel. Global and local statistical analyses and mapping were conducted with ArcGIS version 10.7. In a Sat Scan analysis, a Bernoulli model was employed to identify the most likely spatial clusters of institutional delivery, while, spatial regression was analyzed using ordinary least squares regression and geographically weighted regression to predict hotspot area of institutional delivery. Model performance was assessed using corrected Akaike Information Criteria (AICc) and adjusted $R^2$. The prevalence of institutional delivery in Mozambique was 65.41% (95%CI: 64.20, 66.61), and the spatial distribution of institutional delivery was clustered with global Moran's I = 0.479503. Getis-Ord analysis detected high institutional birth practice among women in Maputo, Maputo City, southwest Inhambane, central Sofala, southern Gaza, and western Niassa regions of Mozambique. Rich wealth index, unwanted pregnancy, vehicle transportation, and presence of skilled birth attendants significantly influenced institutional delivery in geographically weighted regression analysis. In Mozambique, over two-thirds

**Data availability statement:** Third party data was obtained for this study from The DHS Program (https://dhsprogram.com/). Data may be requested from The DHS Program after creating an account and submitting a concept note. More access information can be found on The DHS Program website (https://dhsprogram.com/data/Access-Instructions.cfm). The data set is openly available upon permission from the MEASURE DHS website (https://www.dhsprogram.com/data/available-datasets.cfm). The authors confirm that interested researchers would be able to access these data in the same manner as the authors. The authors also confirm that they had no special access privileges that others would not have.

**Funding:** The author(s) received no specific funding for this work.

**Competing interests:** The authors have declared that no competing interests exist.

of reproductive-age women delivered in health facilities, with high rates in specific regions such as Maputo city, southwest Inhambane, central Sofala, southern Gaza, and western Niassa. Key factors influencing institutional delivery included wealth, unwanted pregnancy, vehicle access, and skilled birth attendants.

## Background

Institutional delivery refers to childbirth within a medical facility under the supervision of trained health professionals, ensuring both mother and child's safety through access to essential facilities and expertise [1]. Utilizing medical institutions for childbirth is a crucial strategy to enhance maternal health. It significantly reduces the risk of maternal deaths and guarantees that deliveries are conducted safely, minimizing complications associated with childbirth [2].

A significant proportion of maternal deaths can be attributed to complications that manifest during labor, childbirth, and the immediate period following birth. The World Health Organization identifies the leading causes of these deaths as severe bleeding, infections, hypertensive disorders (including preeclampsia and eclampsia), and complications during labor and delivery [3]. It is crucial to address these issues with effective healthcare interventions to prevent maternal deaths, which are largely avoidable [4].

The World Health Organization (WHO) targets a world in which "every pregnant woman and newborn receive quality care throughout pregnancy, childbirth, and postnatal period" in order to lower the rate of maternal death [5]. The safe motherhood effort additionally puts a strong emphasis on institutional delivery, which is thought to be able to prevent 16–33% of maternal deaths [6,7], as one component of emergency obstetric care where complex cases can be appropriately handled [8–10].

Globally, WHO estimates that up to 15% of expected births face life-threatening complications during pregnancy, delivery, or postpartum [11]. Even though giving birth at an institution is a crucial strategy of lowering the risks associated with pregnancy and childbirth, many women in developing countries give birth at home [12]. In sub-Saharan Africa, nearly one-third of all childbirths occur without medical supervision or are only overseen by relatives, while traditional birth attendants manage between 23% and 40% of these deliveries [13]. Globally, approximately 830 women die every day from pregnancy and childbirth-related complications [14]. The vast majorities of these fatalities occur in low-income countries and are largely preventable with access to adequate obstetric care [15,16].

In Mozambique, efforts to expand healthcare infrastructure and modern equipment have increased institutional delivery coverage from 71% in 2015 to 87% in 2019 [17]. The government has been supported in formulating and implementing evidence-based policies to improve socio-economic conditions, which can indirectly impact institutional delivery rates [18]. However, there is a notable shortage in critical services such as skilled attendance at birth and the equipment necessary for complex medical procedures [19,20]. These services are costly, imposing a substantial

economic strain on families, particularly in a low-income country like Mozambique where there is minimal health insurance coverage [21–23].

Previous studies have identified several factors significantly associated with institutional delivery, including the sex of the household head, maternal age, occupation, parity, birth order, antenatal care visits [24–27], knowledge of the danger signs of pregnancy and childbirth [28–30], household wealth index [28,31], media exposure [32], maternal education [26,33], prior history of prolonged labor [34], total number of children [34,35], birth preparedness/complication readiness [35,36], and decision-making regarding health care [25,37].

Understanding institutional deliveries in Mozambique requires deeper analysis. To date, no research has utilized geographically weighted regression analysis to explore the factors influencing institutional delivery within the country. Moreover, prior studies have not incorporated spatial regression analysis, which is crucial for addressing location-specific variables. Identifying geographical variability and determinants of institutional deliveries is essential for informed decision-making aimed at increasing institutional delivery rates. This study aimed to fill these gaps by applying spatial analytic techniques to address two key questions: Where are the institutional delivery hotspots within Mozambique? And what factors contribute to spatial disparities in institutional delivery rates across the country? These questions are explored using data from the 2022/23 Mozambique Demographic and Health Survey.

## Methods

### Data source and settings

This study used publicly available data from the 2022/23 Mozambique Demographic and Health Survey (DHS) (S1 Data). The data for the 2022/23 Mozambique Demographic and Health Survey was gathered through a joint effort between the Instituto Nacional de Estatística (INE) and Instituto Nacional de Saúde (INS), with support from USAID's Demographic and Health Surveys (DHS) Program [38]. For the 2022/23 Mozambique DHS, a stratified two-stage sampling design was employed to ensure national and subnational representativeness. In the first stage, 619 enumeration areas (EAs) were selected independently within each stratum 228 from urban areas and 391 from rural areas using probability proportional to size based on the most recent census. In the second stage, a complete household listing was conducted within each selected EA, serving as the sampling frame. Households were then systematically selected using a fixed sampling interval. All women aged 15–49 years who were usual residents or had spent the night before the survey in the selected households were eligible for inclusion. A total 5,642 households and 5,982 women were included in the final analysis.

The dependent and independent variables were extracted from the individual record dataset (IR file). The outcome variable (place of delivery) was recoded as institutional delivery and home delivery from the individual record (IR) data set.

Mozambique consists of ten provinces and Maputo City, which has provincial status. Each province, including Cabo Delgado, Gaza, Inhambane, Manica, Maputo Province, Nampula, Niassa, Sofala, Tete, and Zambezia, is distinct in terms of its geography, cultural background, and economic pursuits. The northern regions such as Cabo Delgado and Niassa are predominantly agricultural with abundant natural resources. In contrast, the southern part of the country, particularly Maputo City and its surrounding province, is characterized by a higher degree of urbanization and notable industrial and commercial activity [39].

### Study variables

**Outcome variable.** "In this study, the independent variable of interest was the 'place of delivery' for births. This variable was categorized into two: 'home delivery', which includes births that occurred at the respondent's own home or another home, and 'institutional delivery', which includes births occurred at the respondent's own home or another home, and 'institutional delivery', which encompasses births that occurred in various healthcare facilities such as central hospitals, provincial or general hospitals, rural or district hospitals, health centers or posts, as well as other public and

private medical settings like private hospitals and clinics" [40–43]. To facilitate spatial analysis, including spatial regression analysis, this categorical variable was transformed into a continuous measure. This was achieved by calculating the weighted proportion of institutional deliveries within each cluster, providing a quantitative measure for subsequent analytical procedures.

**Independent variables.** Maternal education, maternal age, maternal working status, sex of household head, marital status, wealth index, ANC visit, birth order, total children ever born, media exposure, type of pregnancy, history of pregnancy termination, distance to health facility, wanted pregnancy, transportation to the health facility, presence of skilled birth attendant, and place of residence were considered explanatory variables for this study.

## Data management and analysis

Data cleaning and preparation were performed using STATA version 17 and Microsoft Excel, ensuring the dataset's accuracy and readiness for analysis. Following this, spatial analysis was carried out with ArcGIS 10.7 software. Prior to the spatial analysis, the weighted proportions of institutional delivery, which is the outcome variable of interest, along with the independent variables, were computed in STATA. These processed data were then imported into ArcGIS 10.7 to perform the spatial analysis, allowing for the examination of geographical patterns and relationships.

## Spatial analysis

For the purpose of spatial analysis, this study used ArcGIS version 10.7 and Sat-scan version 9.6 to delve into the geographical distribution and patterns of institutional delivery. This analysis included techniques such as spatial autocorrelation, spatial interpolation, and the identification of significant clusters or 'hotspot' areas where institutional deliveries were more common. To determine the pattern of spatial distribution whether it was dispersed, clustered, or random the global Moran's I statistic was employed. This statistical measure is designed to assess spatial autocorrelation across the entire dataset, producing a single value that ranges from -1 to +1. A Moran's I value near -1 suggests a dispersed distribution of institutional deliveries, indicating no specific pattern. Conversely, a value close to +1 suggests a clustered pattern, indicating that institutional deliveries are more likely to occur in certain areas. A value of 0 would indicate a random distribution with no discernible pattern in the geographical occurrence of institutional deliveries [44].

The study employed Getis-Ord Gi* statistics to locate areas with significantly high ('hot-spot') or low ('cold-spot') rates of institutional deliveries. This statistical tool measures spatial autocorrelation, which varies across the location under study, and helps determine the statistical significance of any observed clustering through Z-score and p-value estimates. A high Gi* statistic indicates a hot-spot, signifying a concentration of institutional deliveries, while a low Gi* statistic points to a cold-spot, indicating fewer institutional deliveries in that area. To estimate the prevalence of institutional deliveries among women in areas where data might be missing or sparse, Kriging interpolation techniques were utilized. Among various interpolation methods, Kriging was chosen for this study due to its accuracy, as evidenced by its low residuals and root mean square error, making it a reliable method for predicting values in unmeasured locations [45].

SatScan version 9.6 Software was used to perform a spatial SatScan analysis to identify significant primary and secondary clusters. The primary cluster was defined as the most likely cluster, identified as the scanning window with the maximum likelihood, while secondary clusters referred to other statistically significant clusters detected in the analysis. Since outcome variable was binary, Bernoulli's model was fitted. Women who did not have an institutional delivery were categorized as a control, while those who had were classified as cases. Data for cases, controls, and geographic locations are required for the Bernoulli model. Clusters containing more than the maximum limit were ignored. Both small and significant clusters may be identified using the default maximum spatial cluster size of <50% of the population as an upper limit. The most likely cluster was identified as the scanning window with maximum likelihood, and each cluster was given a p-value by the application of Monte Carlo hypothesis testing [46].

Global Public Health

PLOS

## Spatial regression

Spatial regression analysis is a comprehensive approach that includes both wide-ranging (global) and detailed (local) analysis techniques [47]. We began with the application of global geographical regression models to capture the overall pattern and ensure that the coefficients reflect the diversity across all enumeration areas. This was followed by a more focused local geographical analysis to capture area-specific variations. Exploratory regression was conducted to identify robust models that explain spatial phenomena. It involves evaluating all possible combinations of input candidate explanatory variables to find Ordinary Least Squares (OLS) models that best explain the dependent variable [48].

Ordinary Least Squares (OLS) regression was used to identify factors that influence the geographical variability of institutional deliveries. OLS is a global regression model that uses a single equation to estimate the association between the outcome and the explanatory variables. It relies on the assumption that all of the variables' coefficients are homogeneous and constant throughout the study area [47]. Additionally, the absence of spatial autocorrelation in residuals was verified using the Koenker BP test, which is crucial for determining the appropriateness of geographically weighted regression. To prevent redundancy and ensure that each independent variable contributes uniquely to the model, multicollinearity was assessed using the Variance Inflation Factor (VIF), with a value below 7.5 indicating an acceptable level. A VIF cut-off of 7.5 was selected for this analysis based on recommendations in the literature that suggest a more conservative approach for identifying multicollinearity in Ordinary Least Squares (OLS) regression models [47,49].

The geographically weighted regression was carried out with ArcGIS 10.7 software. The coefficients are not only statistically significant but also logically consistent with expectations, do not duplicate information provided by other variables in the model, and contribute to a strong overall fit as indicated by adjusted R-squared values [50,51]. Despite an insignificant p-value from Koenker's studentized Breusch-Pagan test, the study proceeded with Geographically Weighted Regression (GWR) to explore how the association between dependent and independent variables differs across different locations. GWR considers spatial proximity, assigning distinct regression parameters to each observation in the study area [52]. Model performance was assessed using corrected Akaike Information Criteria (AICc) and adjusted $R^2$. Lower AICc values and higher adjusted $R^2$ indicate better model performance. To incorporate geographic weighting, a spatial kernel was used, and an adaptive kernel was employed due to clustered observations. The AICc was used to determine bandwidth and simplify the model. We used an adaptive kernel because our data showed clustered spatial patterns (such as dense urban vs. sparse rural observations), and the AICc-optimized bandwidth selection confirmed its superior performance [53,54].

## Ethical approval and consent to participate

Since this study is merely a secondary review of the DHS data, ethical approval is not needed. We enrolled with the DHS web archive, requested the dataset for our study, and were granted permission to view and download the data files. As per the DHS study, all participant data were anonymized at the time of survey data collection. Visit in order to understand more about DHS data and ethical standards https://www.dhsprogram.com.

## Results

### Descriptive results

In this study, a total of weighted sample of 5,982 reproductive age women were included. Nearly one-third of women (29.79%) had no formal education. Less than half (46.76%) of women faced significant difficulties accessing healthcare institutions due to distance, and about 4,280 (71.54%) were living in rural areas of Mozambique. About 17.14% of women did not have ANC visits. More than half (52.13%) of women had no access to media (Table 1).

**Table 1.** Descriptive characteristics of study participants in Mozambique, DHS 2022/23.

| Variables | Weighted frequency (n) | Percent (%) |
|---|---|---|
| Sex of household head | | |
| Male | 4,495 | 75.13 |
| Female | 1,488 | 24.87 |
| Maternal age | | |
| 15-24 | 2,712 | 45.32 |
| 25-34 | 2,207 | 36.89 |
| 35-49 | 1,064 | 17.79 |
| Maternal educational level | | |
| No formal education | 1,782 | 29.79 |
| Primary | 2,924 | 48.88 |
| Secondary | 1,202 | 20.08 |
| Higher | 75 | 1.24 |
| Maternal occupational status | | |
| Not working | 4,209 | 70.36 |
| Working | 1,774 | 29.64 |
| Marital status of the mother | | |
| Never married | 336 | 5.62 |
| Currently married | 4,992 | 83.45 |
| Formerly/ever married | 655 | 10.93 |
| Distance to health facility | | |
| Big problem | 2,797 | 46.76 |
| Not a big problem | 3,186 | 53.24 |
| Number of ANC visits | | |
| No visit | 1,026 | 17.14 |
| Had visit | 4,957 | 82.86 |
| Total children ever born | | |
| 1-3 | 3,616 | 60.43 |
| >3 | 2,367 | 39.57 |
| Preceding birth interval | | |
| <24 months | 528 | 8.81 |
| ≥24 months | 5,455 | 91.19 |
| Household wealth index | | |
| Poor | 2,903 | 48.53 |
| Middle | 1,157 | 19.35 |
| Rich | 1,923 | 32.13 |
| Household media exposure | | |
| No | 3,492 | 58.37 |
| Yes | 2,491 | 41.63 |
| Birth order | | |
| First | 1,421 | 23.75 |
| Second | 1,261 | 21.08 |
| Three and more | 3,301 | 55.17 |
| Type of pregnancy | | |
| Single | 5,781 | 96.64 |
| Multiple | 202 | 3.36 |

*(Continued)*

**Table 1.** (Continued)

| Variables | Weighted frequency (n) | Percent (%) |
|---|---|---|
| History of pregnancy termination | | |
| No | 5,477 | 91.55 |
| Yes | 506 | 8.45 |
| Place of residence | | |
| Urban | 1,703 | 28.46 |
| Rural | 4,280 | 71.54 |

### Spatial autocorrelation of institutional delivery in Mozambique

There was significant variation in the institutional deliveries across the regions (Moran's index = 0.479503, p-value <0.001). According to the spatial autocorrelation information, there was a clustering effect in institutional delivery, meaning that there were high institutional deliveries in particular areas and low institutional delivery rates in others. The outputs on both the left and right sides of the panel are equipped with designated keys. The clustered pattern's z-score of 19.167524 indicates that the likelihood of it being a random occurrence is less than 1% (Fig 1).

### Hotspot analysis of institutional delivery in Mozambique

The local Getis-Ord Gi* statistics were employed in the present study to determine institutional delivery hot and cold spots. Significant hot spot areas for institutional deliveries are indicated by the colors red and orange, while cold spot areas are indicated by the color blue. The geographic areas of Maputo, Maputo city, southwest Inhambane, central Sofala, southern Gaza, and western Niassa had the highest rates of institutional delivery. However, the regions with the lowest distribution of institutional delivery were, Tete, Nampula, Zambezia, Cabo Delgado, and Manica (Fig 2).

### Interpolation of institutional delivery in Mozambique

Kriging interpolation identified a high predicted prevalence of institutional delivery in the northwestern (Niassa), central (Manica and Sofala), and southern (Gaza, Maputo Province, and Maputo City) regions of Mozambique, as evidenced by the red to orange shades on the map. In contrast, the northeastern (Cabo Delgado and Nampula) and north-central (Zambezia) regions, along with the western part of Tete Province, exhibit lower predicted institutional delivery rates, as represented by the green shades (Fig 3).

### Sat scan statistical analysis institutional delivery in Mozambique

In the spatial scan statistics, a total of 671 significant clusters were identified; of these, 306 were primary (most likely) clusters. The primary clusters were located in Sofala, Inhambane, Manica, Maputo, and Gaza, centered at (21.731250 S, 32.242984 E) of geographic location with a 543.21 km radius, a relative risk (RR) of 1.42, and a log-likelihood ratio (LLR) of 177.6 at p < 0.001. It showed that women within a spatial window had a 1.42 times higher likelihood of institutional delivery than women outside the spatial window (Fig 4; Table 2).

### Ordinary least square analysis result

The OLS model revealed spatially varying risk factors that affect institutional delivery in Mozambique. This model (Adjusted $R^2$ = 0.387137) explained around 39% of variability in institutional delivery. The Koenker BP test in this study was statistically significant, suggesting that conducting a geographically weighted regression is advisable.

The Joint Wald statistic was statistically significant (p < 0.01), shows that the overall model was significant. Consequently, we performed a geographically weighted regression and obtained the local coefficients for each explanatory variable.

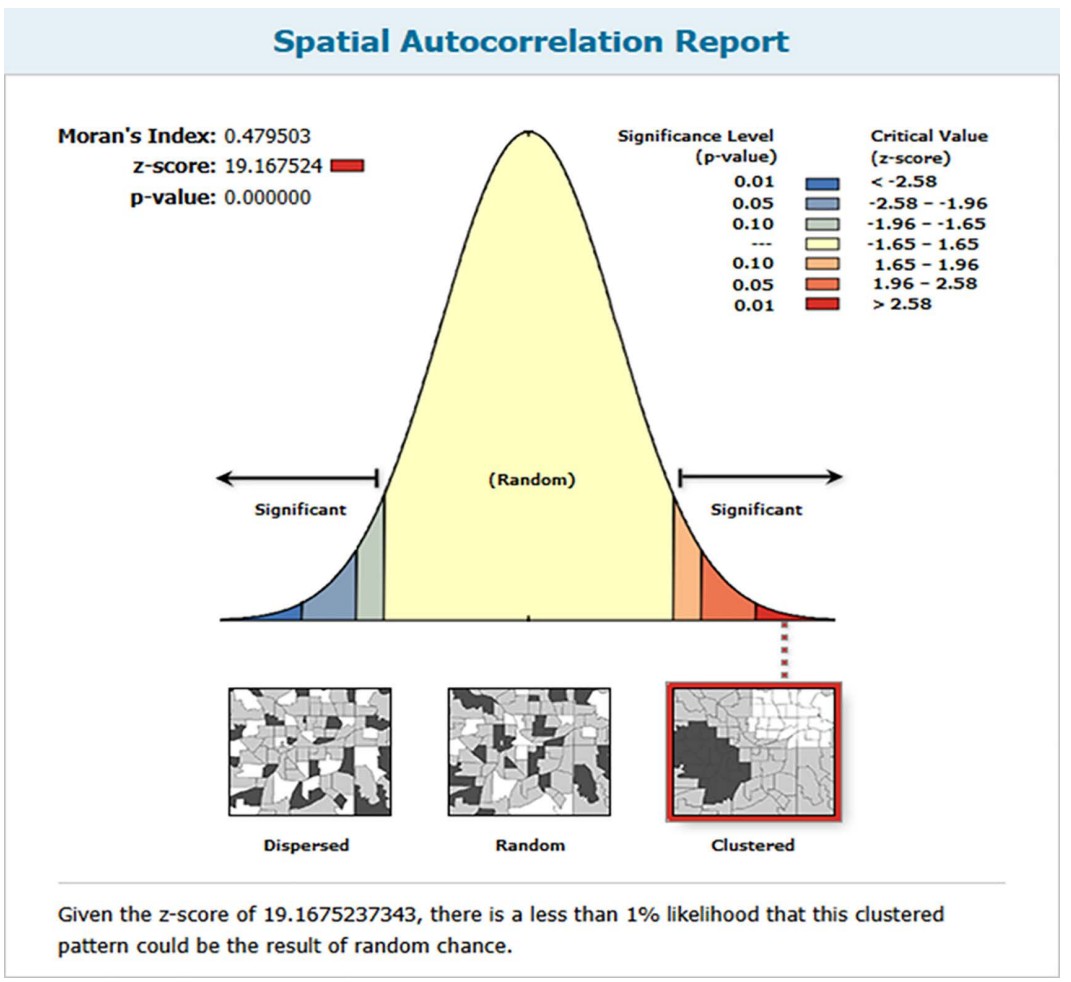

**Fig 1. Spatial autocorrelation of institutional delivery in Mozambique based on feature locations and attribute values using the Global Moran's index statistic, Mozambique DHS 2022/23.**

The proportion of women in the richest wealth index, the proportion of women with unwanted pregnancy, the proportion of women who had vehicle transportation, and the proportion of women who had skilled birth attendant were predictors of institutional delivery hotspot locations (**Table 3**). A unit increase for respondent's wealth index, vehicle transportation, and skilled birth attendant increases institutional delivery by 0.020328, 0.007419, and 0.027000 times respectively.

## Geographic weighted regression analysis

The Geographic weighted regression (GWR) analysis showed that the global model had been improved significantly. The AICc value declined from -29.701in the OLS model to -240.84 the GWR model. In addition, the adjusted R square (0.387) obtained from OLS increased to adjusted R square (0.606), implying that GWR improved the model's ability to predict institutional delivery. Overall, this study found that the GWR analysis outperformed the OLS model (**Table 4**).

In this study GWR showed that the explanatory variables were both strong and weak predictors of institutional delivery. As the proportion of women from rich wealth index increased, the percentage of institutional delivery increased in the entire Cabo Delgade, Niassa, Zambezia, Inhambane, and Tete regions (**Fig 5**). Women who had unwanted current

**Fig 2. Hotspot analysis of institutional delivery in Mozambique based on feature locations and attribute values using the Global Moran's index statistic, Mozambique DHS 2022/23.** Source shape file: https://data.humdata.org/dataset/5e8d83a5-1210-49be-b7d9-cf286dbc15df.

pregnancy had a strong negative relationship with institutional delivery. As the proportion of women who had unwanted current pregnancy increased, the existence of institutional delivery in Nampula and Zambezia regions decreased (Fig 6). As shown in (Fig 7), higher beta coefficients of vehicle transport indicating a stronger association with institutional delivery were observed in the regions of Cabo Delgado, Tete, and Inhambane. As the proportion of women with skilled birth attendant increased, the existence of institutional delivery in Tete, Niassa, and Cabo Delgade regions increased (Fig 8).

## Discussion

Institutional delivery is a critical factor in reducing maternal and neonatal morbidity and mortality, as it ensures skilled birth attendance and access to emergency obstetric care [55]. Despite global efforts, institutional delivery services remain underutilized in many low- and middle-income countries due to socio-demographic, economic, and cultural factors [56,57].

In this study, the prevalence of institutional delivery in Mozambique among reproductive-age women was 65.41% (95%CI: 64.20, 66.61) and ranged from 2.21% in the Cidade de Maputo region to 26.20% in the Napula region. The finding was consistent with the study conducted in Uganda, 65.29% [58]. This was lower than a study reported in Southwest

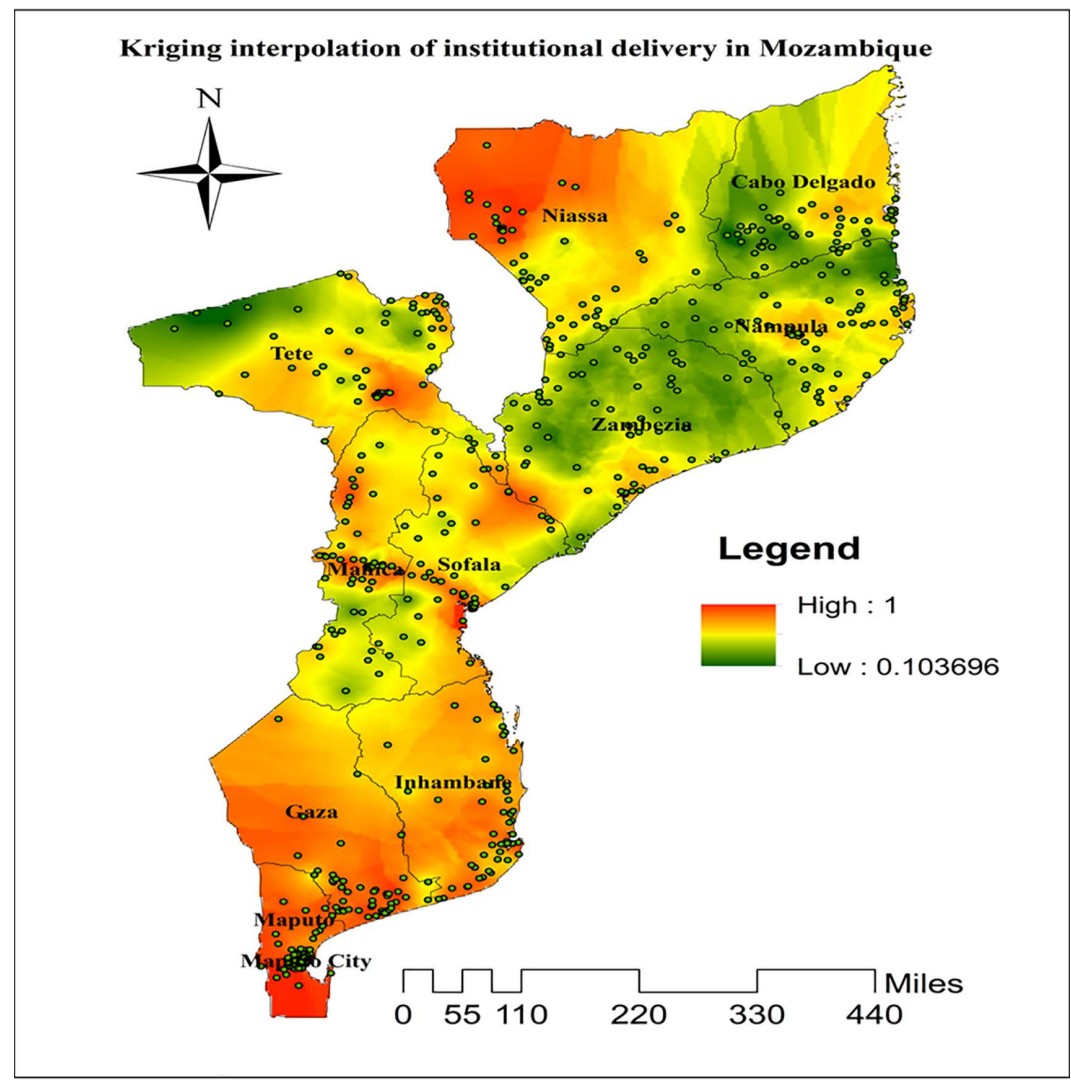

**Fig 3. Kriging interpolation analysis of institutional delivery in Mozambique, DHS 2022/23.** Source shape file: https://data.humdata.org/dataset/5e8d83a5-1210-49be-b7d9-cf286dbc15df.

Ethiopia [59,60] and Tanzania, 74.64% [58]. On the other hand, the prevalence of institutional delivery in this study was higher than the findings conducted Kenya, 62.10% [61].

The observed prevalence of institutional delivery in Mozambique (65.41%) can be attributed to several factors that differentiate it from other regions, such as Southwest Ethiopia, Tanzania, and Kenya. In Mozambique, barriers like geographical accessibility, socioeconomic disparities, cultural beliefs, and limited availability of skilled health personnel contribute to a lower institutional delivery rate compared to Tanzania, where recent healthcare reforms and improved maternal health programs have enhanced access to delivery services [62]. Conversely, Mozambique's prevalence is higher than Kenya's, which may be due to Mozambique's concerted efforts to expand maternal health services, enhance community health education, and integrate maternal healthcare into broader public health initiatives, despite ongoing challenges [63].

This study also showed that institutional delivery was clustered spatially at the enumeration area level. Getis-Ord spatial analysis showed that hot spot and cold spot enumeration areas were detected using cluster outlier analysis. This

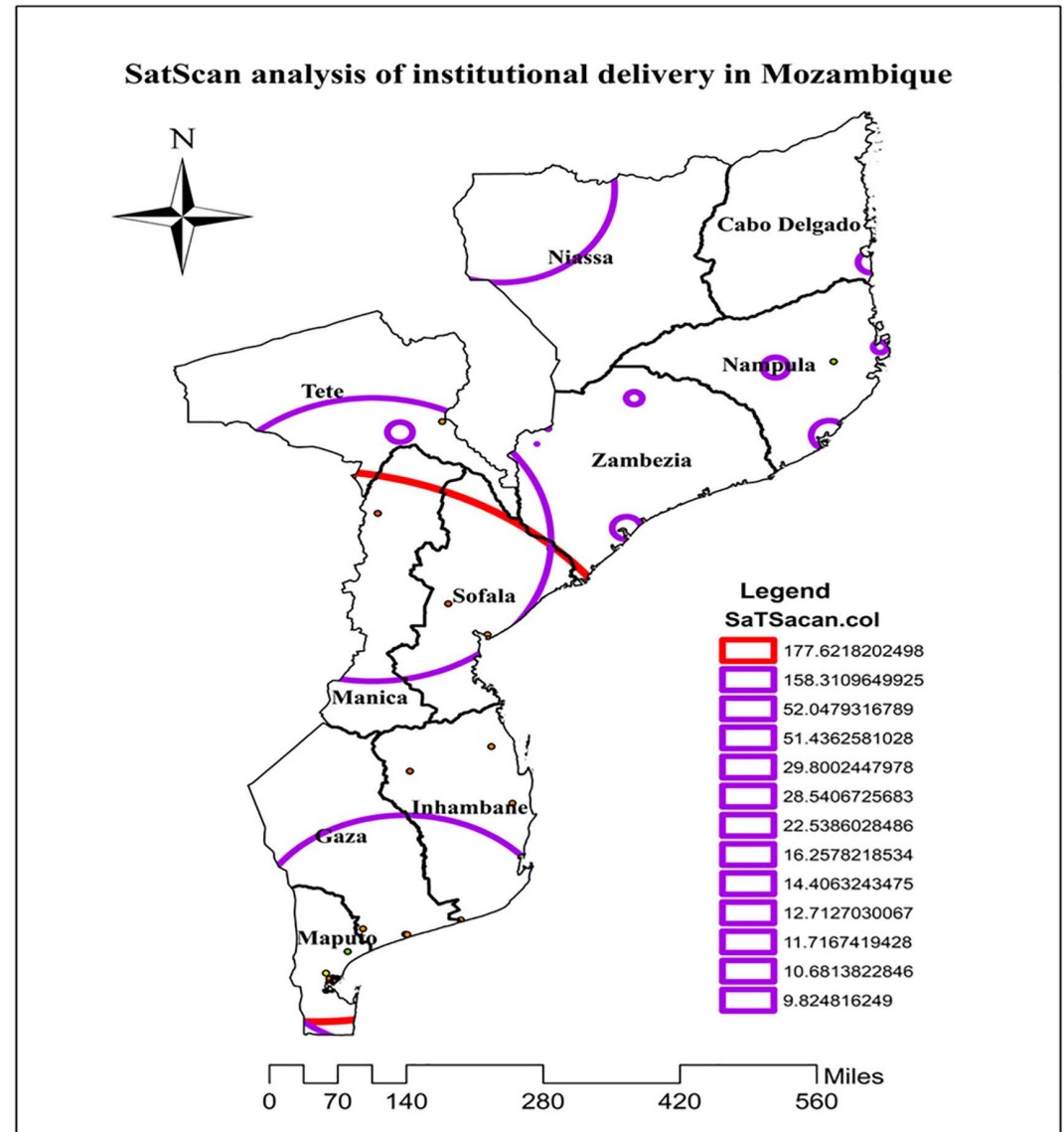

**Fig 4. Sat scan analysis of institutional delivery in Mozambique, DHS 2022/23.** Source shape file: https://data.humdata.org/dataset/5e8d83a5-1210-49be-b7d9-cf286dbc15df.

finding aligns with other studies that have highlighted the presence of geographical clustering of institutional deliveries in Mozambique [64,65]. The most plausible explanation for this spatial variation could be the geographical heterogeneity of Mozambique, including differences in topography, infrastructure, and access to health facilities, which affect the availability and utilization of healthcare services. Rural and remote areas often face significant barriers such as poor road networks, limited transportation options, and inadequate healthcare infrastructure, contributing to lower institutional delivery rates [66]. Mozambique exhibits significant spatial variations in institutional delivery due to several interrelated factors, including geographical heterogeneity, disparities in healthcare infrastructure, socio-economic differences, and cultural influences. The country has a diverse topography, with remote and rural areas facing greater barriers to accessing health facilities compared to urban regions, where institutional delivery rates tend to be higher. Poor road networks and limited

**Table 2. Sat scan analysis of institutional delivery in Mozambique: DHS 2022/23.**

| Cluster | Enumeration areas | Coordinates/radius | Population (n) | Cases (n) | RR | LLR | P-value |
|---|---|---|---|---|---|---|---|
| 1(306) | 505, 339, 345, 344, 346, 492, 446, 338, 336, 347, 337, 348, 499, 349, 396, 393, 432, 352, 351, 394, 433, 395, 498, 506, 431, 441, 353, 385, 325, 356, 488, 350, 403, 357, 383, 355, 495, 461, 440, 494, 420, 497, 459, 429, 316, 308, 450, 315, 482, 428, 324, 323, 310, 314, 341, 317, 480, 309, 322, 478, 313, 384, 386, 311, 312, 342, 479, 358, 496, 457, 360, 321, 331, 462, 332, 320, 412, 460, 330, 493, 382, 452, 487, 343, 365, 414, 366, 361, 481, 364, 362, 367, 397, 363, 453, 368, 413, 371, 373, 370, 551, 377, 504, 372, 369, 411, 378, 379, 476, 374, 449, 477, 475, 410, 375, 451, 484, 489, 491, 555, 550, 485, 398, 421, 500, 422, 447, 456, 490, 509, 359, 501, 473, 455, 486, 472, 448, 508, 474, 507, 434, 510, 553, 516, 436, 513, 514, 458, 326, 427, 454, 438, 425, 515, 467, 423, 424, 469, 426, 512, 399, 468, 435, 511, 470, 471, 483, 415, 556, 418, 417, 465, 502, 445, 409, 416, 543, 503, 400, 444, 466, 568, 552, 437, 464, 463, 419, 439, 442, 319, 443, 554, 318, 401, 544, 327, 567, 566, 563, 558, 522, 402, 527, 559, 340, 534, 557, 539, 561, 525, 538, 406, 562, 560, 616, 526, 546, 533, 328, 615, 614, 617, 530, 589, 610, 531, 523, 535, 590, 611, 598, 519, 537, 599, 536, 532, 529, 605, 524, 521, 528, 596, 602, 604, 612, 597, 613, 593, 603, 606, 595, 600, 592, 601, 591, 392, 518, 594, 609, 607, 608, 587, 584, 575, 586, 581, 582, 547, 583, 520, 578, 580, 588, 577, 579, 576, 585, 329, 517, 572, 570, 571, 549, 548, 545, 541, 619, 618, 404, 540, 542, 334, 565, 569, 564, 405, 333, 388, 407, 275, 408, 381, 380, 376, 335, 387 | (21.731250 S, 32.242984 E)/543.21 km | 1732 | 1434 | 1.42 | 177.6 | <0.0001 |
| 2(177) | 467, 468, 471, 469, 470, 512, 483, 515, 507, 508, 514, 511, 509, 513, 490, 510, 489, 491, 475, 477, 476, 516, 500, 486, 474, 487, 473, 484, 472, 481, 485, 502, 553, 504, 493, 555, 496, 503, 501, 479, 556, 466, 482, 478, 480, 543, 497, 552, 550, 465, 554, 494, 495, 563, 463, 464, 557, 558, 567, 562, 559, 560, 561, 522, 551, 589, 615, 616, 590, 610, 565, 617, 599, 598, 614, 525, 611, 526, 592, 498, 593, 591, 602, 596, 603, 527, 534, 597, 595, 587, 566, 601, 600, 585, 594, 583, 605, 523, 586, 604, 612, 588, 529, 582, 613, 539, 584, 581, 533, 606, 530, 437, 535, 528, 575, 572, 488, 524, 578, 571, 580, 536, 608, 579, 607, 570, 538, 531, 576, 577, 609, 458, 568, 532, 537, 438, 419, 518, 618, 521, 544, 520, 519, 546, 517, 506, 619, 549, 547, 541, 548, 444, 439, 540, 436, 542, 564, 445, 569, 545, 434, 435, 442, 427, 499, 423, 456, 426, 443, 425, 424, 415, 418, 417, 431, 416, 454 | (25.035391 S, 33.641761 E)/230.95km | 762 | 696 | 1.49 | 158.3 | <0.001 |
| 3(20) | 22, 12, 24, 23, 53, 38, 48, 54, 17, 37, 3, 2, 4, 1, 5, 6, 7, 25, 16, 26 | (11.856899 S, 35.056739 E)/184.02km | 191 | 182 | 1.48 | 52.1 | <0.001 |
| 4(126) | 319, 318, 327, 326, 328, 340, 329, 359, 334, 343, 400, 342, 333, 322, 323, 341, 358, 324, 360, 331, 311, 310, 399, 309, 312, 313, 314, 308, 320, 317, 315, 316, 321, 335, 275, 404, 330, 406, 405, 350, 353, 357, 332, 401, 325, 355, 410, 402, 412, 356, 391, 411, 382, 388, 392, 414, 354, 413, 383, 276, 352, 398, 351, 390, 389, 257, 262, 256, 249, 250, 251, 305, 255, 295, 253, 384, 252, 254, 397, 261, 260, 307, 297, 376, 387, 296, 385, 379, 348, 377, 349, 378, 306, 347, 291, 375, 292, 370, 273, 372, 374, 369, 282, 371, 394, 368, 366, 367, 373, 361, 365, 362, 227, 364, 363, 393, 225, 259, 195, 346, 344, 409, 386, 274, 229, 408 | (18.069717 S, 33.164488 E)/279.13km | 1058 | 829 | 1.25 | 51.4 | <0.001 |
| 5(11) | 249, 251, 250, 256, 257, 253, 255, 252, 254, 260, 261 | (16.165541 S, 33.569959 E)/19.79 km | 96 | 93 | 1.50 | 29.8 | <0.001 |
| 6(8) | 175, 120, 114, 119, 113, 118, 115, 117 | (15.019698 S, 39.130171 E)/20.16 km | 145 | 133 | 1.42 | 28.5 | <0.001 |
| 7(9) | 93, 111, 59, 58, 62, 55, 56, 57, 61 | (13.152397 S, 40.545217 E)/20.76 km | 96 | 90 | 1.45 | 22.5 | <0.001 |

*(Continued)*

**Table 2.** (Continued)

| Cluster | Enumeration areas | Coordinates/radius | Population (n) | Cases (n) | RR | LLR | P-value |
|---|---|---|---|---|---|---|---|
| 8(1) | 191 | (16.104489 S, 35.764213 E)/ 0 km | 38 | 38 | 1.54 | 16.3 | <0.0001 |
| 9(2) | 133, 131 | (14.657717 S, 40.680162 E)/10.93 km | 68 | 63 | 1.43 | 14.4 | <0.001 |
| 10(2) | 123, 136 | (16.220412 S, 39.930296 E)/28.87 km | 46 | 44 | 1.47 | 12.7 | 0.0014 |
| 11(6) | 183, 184, 185, 237, 197, 198 | (17.864946 S, 36.924892 E)/22.44 km | 55 | 51 | 1.43 | 11.7 | 0.0037 |
| 12(2) | 187, 189 | (15.562901 S, 37.038216 E)/12.64 km | 25 | 25 | 1.53 | 10.7 | 0.012 |
| 13(1) | 222 | (16.375074 S, 35.598072 E)/0 km | 23 | 23 | 1.53 | 9.8 | 0.025 |

**Table 3. Summary of ordinary list square result.**

| Variable | Coefficient | Standard error | t-statistic | Probability | Robust SE | Robust t-statistics | Robust probability | VIF |
|---|---|---|---|---|---|---|---|---|
| Intercept | 0.735427 | 0.015455 | 47.58395 | <0.001 | 0.01888 | 38.943706 | <0.001 | |
| Rich wealth index | 0.020328 | 0.004529 | 4.488919 | <0.001 | 0.00479 | 4.235732 | <0.001 | 5.27 |
| Unwanted pregnancy | −0.027621 | 0.002609 | −10.5874 | <0.001 | 0.00443 | −6.230969 | <0.001 | 6.61 |
| vehicle transportation | 0.007419 | 0.002720 | 2.727964 | <0.001 | 0.00298 | 2.493699 | <0.001 | 4.98 |
| Skilled birth attendant | 0.027000 | 0.003274 | 8.246302 | <0.001 | 0.00395 | 6.838417 | <0.001 | 1.98 |
| OLS Diagnosis | | | | | | | | |
| Number of observations | 612 | | | Akaike's Information Criterion (AICc) | | −29.705051 | | |
| Multiple R-Squared | 0.387167 | | | Adjusted R-Squared | | 0.3871370 | | |
| Joint F-Statistic | 39.596001 | | | Prob(>F), (10,601) degrees of freedom | | <0.001 | | |
| Joint Wald Statistic | 172.741013 | | | Prob(>chi-squared), (10) degrees of freedom | | <0.001 | | |
| Koenker (BP) Statistic | 131.187484 | | | Prob(>chi-squared), (10) degrees of freedom | | <0.001 | | |
| Jarque-Bera Statistic | 50.539723 | | | Prob(>chi-squared), (2) degrees of freedom | | <0.001 | | |

**Table 4. Geographic weighted regression model for the institutional delivery in Mozambique.**

| Explanatory variables | Rich wealth index, unwanted pregnancy, vehicle transportation, Skilled birth attendant |
|---|---|
| Bandwidth | 108282.41 |
| Residual squares | 212.58 |
| Effective number | 124.15 |
| Sigma | 0.66 |
| Akaike's Information Criterion (AICc) | −240.84 |
| Multiple R-Squared | 0.676 |
| Adjusted R-square | 0.606 |

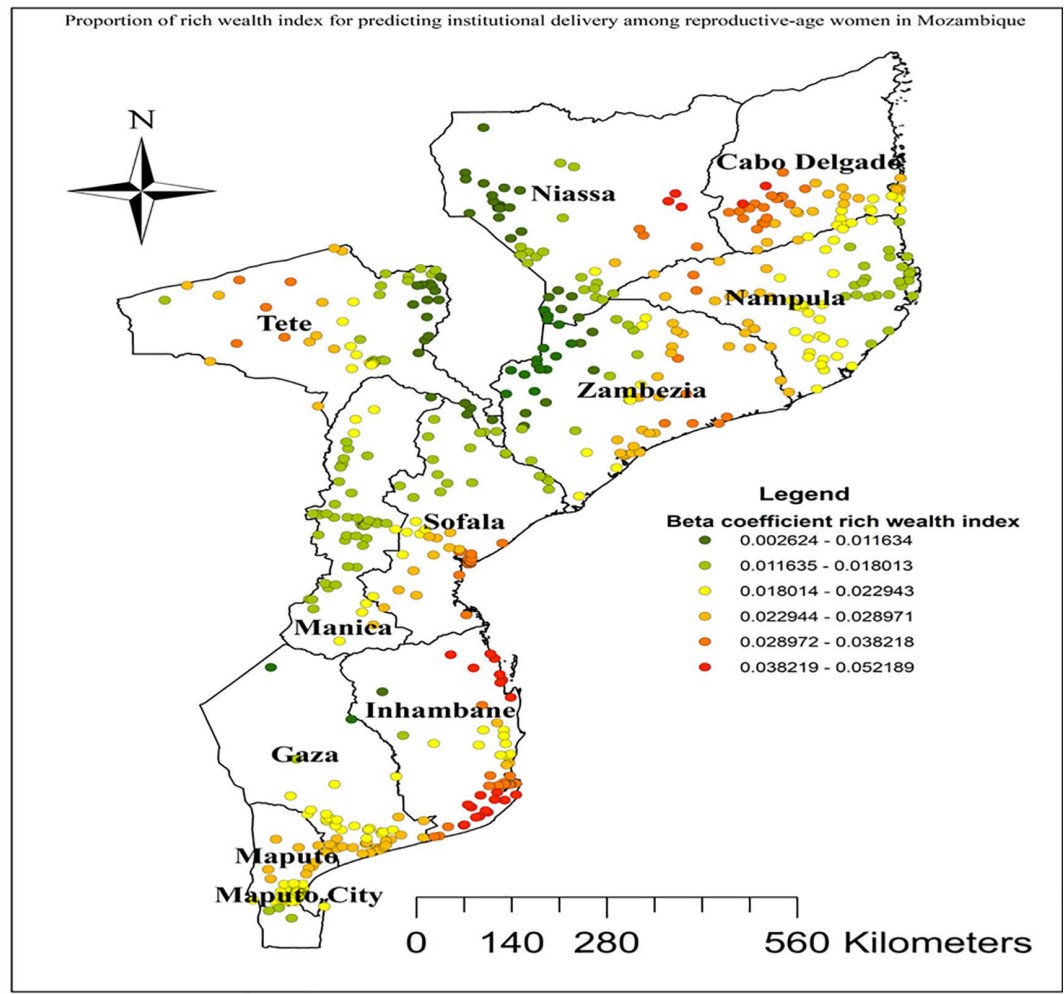

**Fig 5. Women coming from rich wealth index household GWR coefficients for predicting institutional delivery among reproductive age women in Mozambique, DHS 2022/23.** Source shape file: https://data.humdata.org/dataset/5e8d83a5-1210-49be-b7d9-cf286dbc15df.

transportation options in some regions further exacerbate these disparities, making it difficult for pregnant women to reach health facilities in a timely manner [67].

In the spatial regression analysis, rich wealth index, unwanted pregnancy, vehicle transportation, and presence skilled birth attendant were significant predictors of hotspot areas of institutional delivery among reproductive age women. Institutional delivery was positively correlated with the rich wealth quantiles. Geographical areas identified for higher coefficients of women with rich wealth quantiles in the entire Cabo Delgade, Niassa, Zambezia, Inhambane, and Tete regions were fitted with hot spots areas of institutional delivery. This finding is supported by studies conducted in India [68]. This correlation can be attributed to the higher financial capacity of women in rich wealth quantiles, enabling them to afford healthcare services, transportation, and other associated costs, which are often barriers for poorer women. Moreover, wealthier women are generally more educated and informed about the benefits of institutional delivery [69,70].

Geographical areas with higher coefficients of unwanted pregnancies in Nampula and Zambezia regions were identified as hot spots with lower rates of institutional delivery. This finding is supported by a studies conducted in Nigeria [71] and

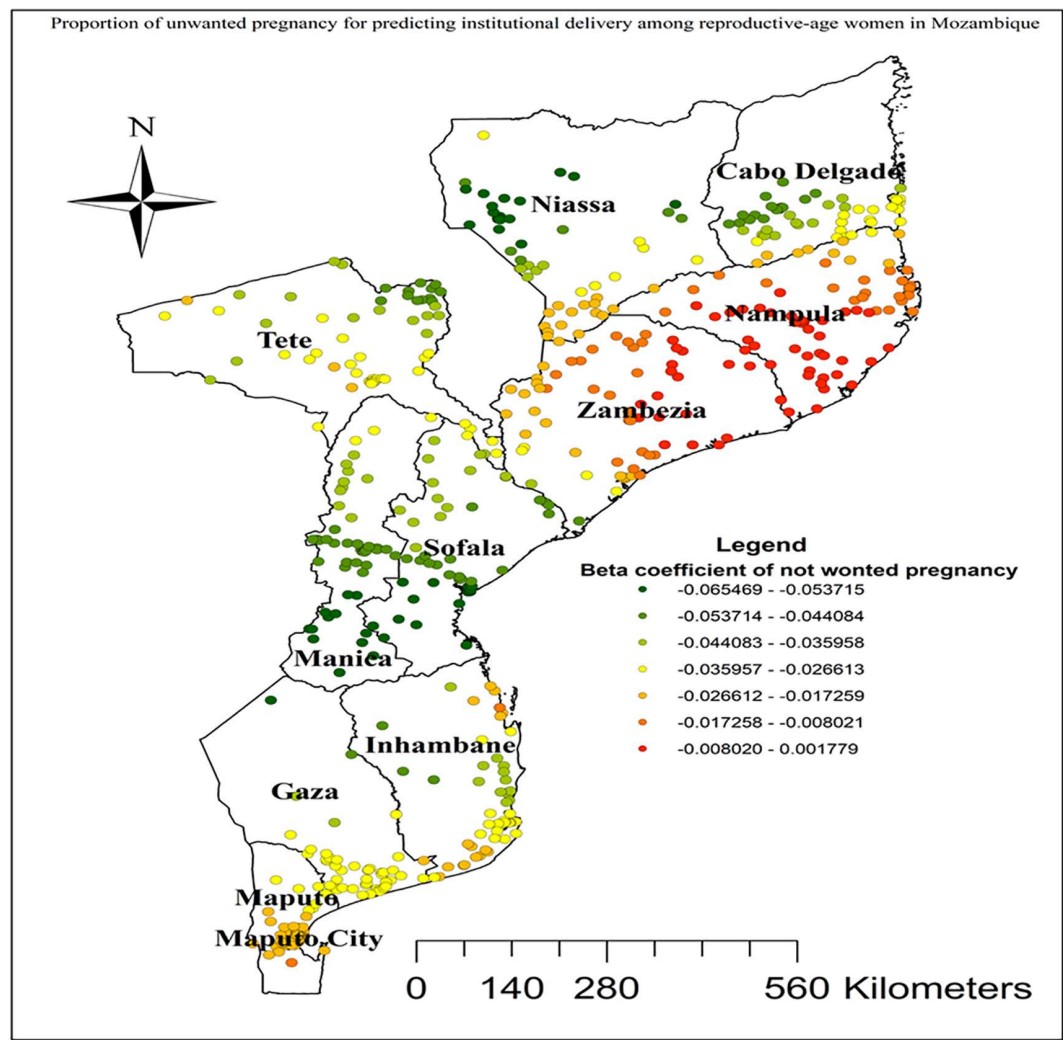

**Fig 6. Unwanted pregnancy GWR coefficients for predicting institutional delivery among reproductive age women in Mozambique, DHS 2022/23.** Source shape file: https://data.humdata.org/dataset/5e8d83a5-1210-49be-b7d9-cf286dbc15df.

Bangladesh [72]. Geographical areas with higher coefficients of unwanted pregnancies are often identified as hot spots with lower rates of institutional delivery due to several intertwined factors. Unwanted pregnancies frequently correlate with inadequate access to reproductive health services, lower socioeconomic status, and limited educational opportunities, which can also impact the likelihood of institutional delivery. Women in these areas may lack awareness or resources for prenatal care and institutional delivery options, leading to a reliance on informal or home-based childbirth practices. Additionally, the stigma and lack of supportive infrastructure for unwanted pregnancies can further deter women from seeking institutional delivery [73,74].

Increased occurrence of institutional deliveries was also observed among women who had vehicle transportation in Cabo Delgade, Tete, and Inhambane regions. This finding can be explained by the fact that vehicle transportation significantly enhances access to healthcare facilities, facilitating timely and reliable access to institutional delivery services. The availability of transportation likely reduces barriers related to distance and travel time, enabling more women to reach healthcare facilities where they can receive comprehensive and professional delivery care [75,76]. Moreover,

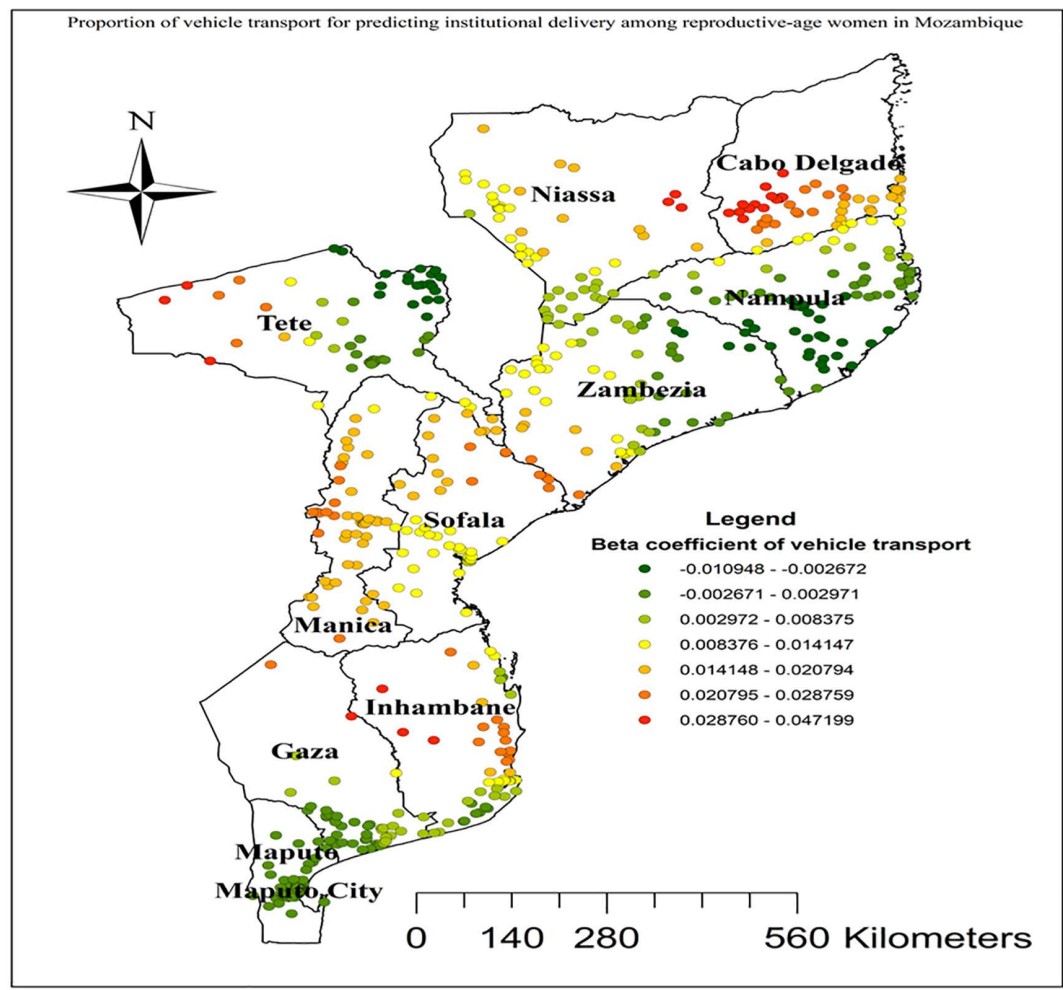

**Fig 7. Vehicle transport GWR coefficients for predicting institutional delivery among reproductive age women in Mozambique, DHS 2022/23.** Source shape file: https://data.humdata.org/dataset/5e8d83a5-1210-49be-b7d9-cf286dbc15df.

geographical areas identified for higher coefficients of women with the presence of skilled birth attendance in Tete, Niassa, and Cabo Delgade regions were positively correlated with hot spots areas of institutional delivery. This finding is supported by studies conducted in Haiti [77] and Ghana [78]. The positive correlation between geographical areas identified for higher coefficients of women with the presence of skilled birth attendance and hot spots of institutional delivery in Mozambique can be attributed to several factors. First, areas with better access to skilled birth attendance often have improved healthcare infrastructure, including well-equipped health facilities and trained personnel, which also promote institutional deliveries [79].

Additionally, these areas may have targeted health interventions, such as community education and outreach programs, that emphasize the benefits of institutional delivery, further encouraging women to seek care at health facilities [80]. Socioeconomic factors, such as higher education levels and income, prevalent in these regions, also play a role, as they are associated with increased healthcare-seeking behavior, including both skilled birth attendance and institutional delivery [81].

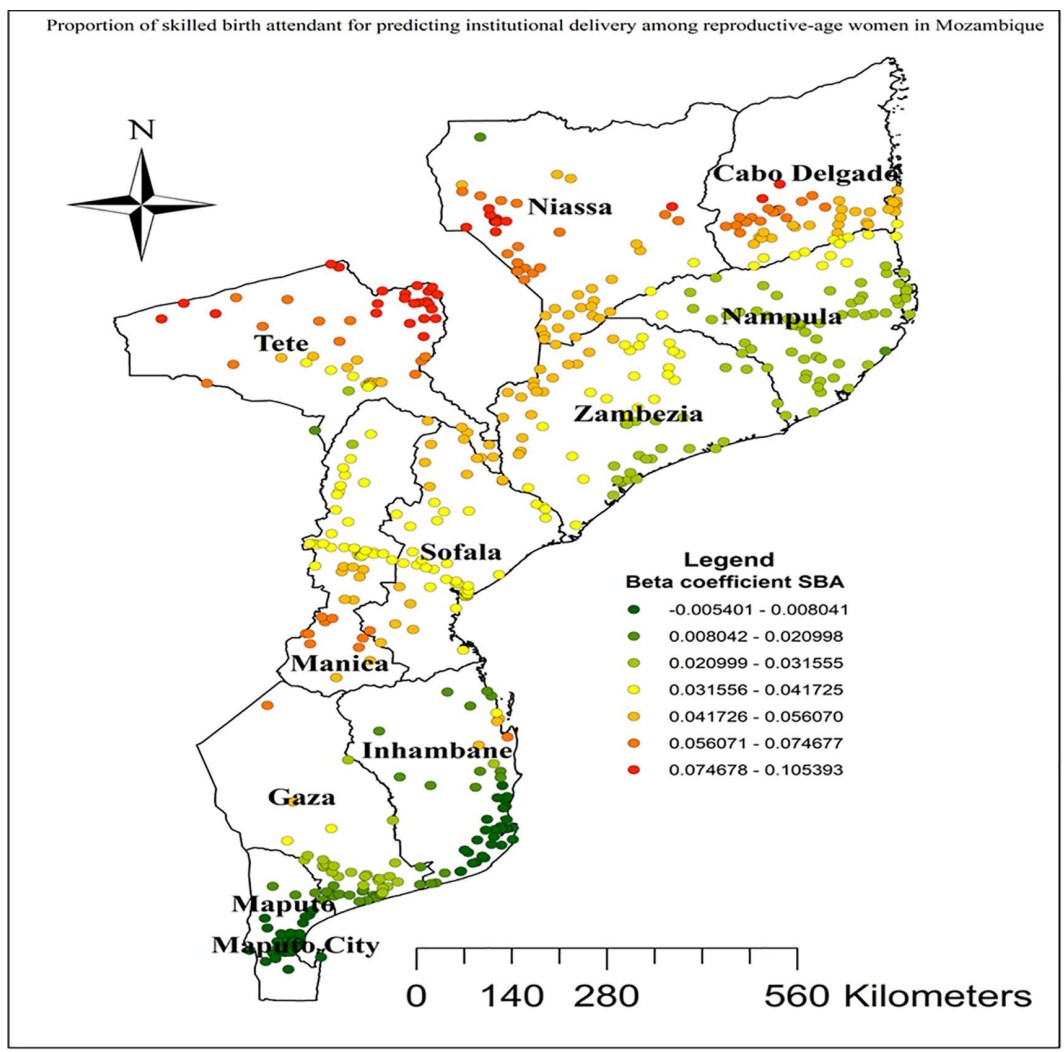

**Fig 8. Skilled birth attendant GWR coefficients for predicting institutional delivery among reproductive age women in Mozambique, DHS 2022/23.** Source shape file: https://data.humdata.org/dataset/5e8d83a5-1210-49be-b7d9-cf286dbc15df.

## Strength and limitations of the study

This study utilized the latest nationally representative MDHS data, gathered using standardized and validated instruments. Geographic Information System (GIS) and Sat Scan statistical tests were employed to identify significant clusters of institutional delivery. Additionally, geographic weighted regression analysis highlighted the impact of predictors in specific areas. A limitation of the study was the difficulty in pinpointing exact case locations, as location data were shifted by 2 km in urban areas and 5–10 km in rural areas to maintain confidentiality.

## Conclusions

In the regions of Mozambique, more than two-third of reproductive age women gave birth at health institution. This study showed that the distribution of institutional delivery was clustered at the enumeration area level in the emerging region of Mozambique. Hotspot regions for institutional delivery were identified in Maputo City, southwest Inhambane, central

Sofala, southern Gaza, and western Niassa. Spatial regression analysis identified wealth index, unwanted pregnancy, vehicle transportation, and the presence of skilled birth attendants as significant determinants of institutional delivery in Mozambique. To improve hospital birth rates, targeted interventions should focus on increasing healthcare accessibility in rural and underserved regions by expanding healthcare infrastructure and transportation services. Strengthening community awareness programs on the benefits of skilled birth attendance and institutional delivery is crucial.

## Supporting information

**S1 Data. Dataset used for the analysis of institutional delivery and its determinants among reproductive-age women in Mozambique: A Geographic weighted regression.** Source: Mozambique Demographic and Health Survey (DHS) 2022/23.
(XLS)

## Author contributions

**Conceptualization:** Alebachew Ferede Zegeye, Solomon Gedlu Nigatu.

**Data curation:** Alebachew Ferede Zegeye, Alemu Teshale, Mequanint Kassa, Solomon Gedlu Nigatu.

**Formal analysis:** Alebachew Ferede Zegeye, Habtu Kifle Negash, Alemu Teshale, Mequanint Kassa, Woretaww Endalew, Araya Mesfin Nigatu.

**Funding acquisition:** Belete Assefa.

**Investigation:** Alebachew Ferede Zegeye.

**Methodology:** Alebachew Ferede Zegeye, Woretaww Endalew, Belete Assefa, Araya Mesfin Nigatu, Solomon Gedlu Nigatu.

**Resources:** Woretaww Endalew.

**Software:** Alebachew Ferede Zegeye, Habtu Kifle Negash, Alemu Teshale, Mequanint Kassa, Araya Mesfin Nigatu, Solomon Gedlu Nigatu.

**Supervision:** Araya Mesfin Nigatu, Solomon Gedlu Nigatu.

**Validation:** Habtu Kifle Negash, Woretaww Endalew.

**Visualization:** Belete Assefa.

**Writing – original draft:** Alebachew Ferede Zegeye, Habtu Kifle Negash, Alemu Teshale, Mequanint Kassa, Woretaww Endalew, Belete Assefa, Solomon Gedlu Nigatu.

**Writing – review & editing:** Alebachew Ferede Zegeye, Habtu Kifle Negash, Araya Mesfin Nigatu, Solomon Gedlu Nigatu.

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
