## [Decision Letter · Decision Letter 0]

17 Mar 2025

PGPH-D-24-03063

Institutional delivery and its determinants among reproductive-age women in Mozambique: A Geographic weighted regression

Dear Dr. Zegeye,

Thank you for submitting your manuscript to PLOS Global Public Health. After careful consideration, we feel that it has merit but does not fully meet PLOS Global Public Health’s publication criteria as it currently stands. Therefore, we invite you to submit a revised version of the manuscript that addresses the points raised during the review process.

We look forward to receiving your revised manuscript.

Kind regards,

Abhinav Sinha, M.D.

Academic Editor

Journal Requirements:

 1. We have amended your Competing Interest statement and Funding Disclosure to comply with journal style. We kindly ask that you double check the statement and let us know if anything is incorrect.  2. Please send separate figure files in .tif or .eps format. Also, remove the figures from your manuscript file but keep the legends. For more information about figure files please see our guidelines:  https://journals.plos.org/globalpublichealth/s/figures https://journals.plos.org/globalpublichealth/s/figures#loc-file-requirements   3. Thank you for uploading your study's underlying data set. Unfortunately, the repository you have noted in your Data Availability statement does not qualify as an acceptable data repository according to PLOS's standards.  At this time, please upload the minimal data set necessary to replicate your study's findings to a stable, public repository (such as figshare or Dryad) and provide us with the relevant URLs, DOIs, or accession numbers that may be used to access these data. For a list of recommended repositories and additional information on PLOS standards for data deposition, please see https://journals.plos.org/plosone/s/recommended-repositories.  4. Some material included in your submission may be copyrighted. According to PLOS’s copyright policy, authors who use figures or other material (e.g., graphics, clipart, maps) from another author or copyright holder must demonstrate or obtain permission to publish this material under the Creative Commons Attribution 4.0 International (CC BY 4.0) License used by PLOS journals. Please closely review the details of PLOS’s copyright requirements here: PLOS Licenses and Copyright. If you need to request permissions from a copyright holder, you may use PLOS's Copyright Content Permission form. Please respond directly to this email or email the journal office and provide any known details concerning your material's license terms and permissions required for reuse, even if you have not yet obtained copyright permissions or are unsure of your material's copyright compatibility.  Potential Copyright Issues: a. “Figures 2, 3, 4, 5, 6, 7, 8”: please (a) provide a direct link to the base layer of the map (i.e., the country or region border shape) and ensure this is also included in the figure legend; and (b) provide a link to the terms of use / license information for the base layer image or shapefile. We cannot publish proprietary or copyrighted maps (e.g. Google Maps, Mapquest) and the terms of use for your map base layer must be compatible with our CC-BY 4.0 license.  Note: if you created the map in a software program like R or ArcGIS, please locate and indicate the source of the basemap shapefile onto which data has been plotted. If your map was obtained from a copyrighted source please amend the figure so that the base map used is from an openly available source. Alternatively, please provide explicit written permission from the copyright holder granting you the right to publish the material under our CC-BY 4.0 license. Please note that the following CC BY licenses are compatible with PLOS license: CC BY 4.0, CC BY 2.0 and CC BY 3.0, meanwhile such licenses as CC BY-ND 3.0 and others are not compatible due to additional restrictions.  If you are unsure whether you can use a map or not, please do reach out and we will be able to help you. The following websites are good examples of where you can source open access or public domain maps: * U.S. Geological Survey (USGS) - All maps are in the public domain. (http://www.usgs.gov) * PlaniGlobe - All maps are published under a Creative Commons license so please cite “PlaniGlobe, http://www.planiglobe.com, CC BY 2.0” in the image credit after the caption. (http://www.planiglobe.com/?lang=enl) * Natural Earth - All maps are public domain. (http://www.naturalearthdata.com/about/terms-of-use/)

Additional Editor Comments (if provided):

Reviewers' comments:

Reviewer's Responses to Questions

**Comments to the Author**

1. Does this manuscript meet PLOS Global Public Health’s publication criteria?

Reviewer #1: Yes

Reviewer #2: Yes

2. Has the statistical analysis been performed appropriately and rigorously?

Reviewer #1: Yes

Reviewer #2: Yes

3. Have the authors made all data underlying the findings in their manuscript fully available (please refer to the Data Availability Statement at the start of the manuscript PDF file)?

Reviewer #1: Yes

Reviewer #2: No

4. Is the manuscript presented in an intelligible fashion and written in standard English?

Reviewer #1: Yes

Reviewer #2: Yes

Reviewer #1: 1. Line 115-119: Sampling process to explicitly mention how a figure of 619 enumeration areas was decided. What was the sampling frame and the sampling interval for the selection of households? Were any inclusion and exclusion criteria used for selecting women from the households? Total number of household / women whose data were analyzed must be mentioned in the methodology.

2. Line 179: Operational definitions of primary and secondary clusters to be mentioned in the methodology.

3. Line 204: What was the reason behind selecting the cut-off of 7.5 for VIF? Generally accepted values are either 5 or 10. Kindly explain and include it in the manuscript.

4. Line 209: Full form of GWR to be mentioned first and then the abbreviation to be mentioned in the parenthesis.

5. Line 221: Avoid using terms like ‘big problem’ in the results unless some objective criteria were used for quantification of the problem. Any cut off distance was used for this classification? If not, rephrase the sentence appropriately.

6. Line 224: Table 1 – Sex of the household head- ‘meal’ to be replaced with “male”.

7. Line 250-251: High prevalence was reported in northwestern part of the country (as evidenced from the map: refer to Niassa province) and not northeastern as mentioned in the text! Check and modify accordingly.

8. Line 273: Is the last word of this line ‘high’ required? If not, omit.

9. Line 279: Replace the word ‘vary’ with ‘varying’.

10. Line 291: Table 3 – For the ease of comprehension it would be better to include only the value of the coefficient and the p value for the variables in the table. The other statistics may be omitted.

11. Line 296: ‘R2’ to be replaced with R square .

12. Line 305 – 306: Check the sentence – As shown in Figure 7 …it appears incoherent – modify appropriately.

13. Line 307: Replace the word ‘whose’ with word ‘with’.

14. Line 446: References poorly compiled, use standard Vancouvre style

Ref 1: Replace Organization WH with World Health Organization..

Ref 3: replace word ‘retrieved’ with ‘cited on’.

Ref 4: check page number; it can’t be 1 to 0, if it is 1 to 10, it must be written as such.

Ref 5: Rewrite ‘Bjog’ as per the standard format.

Reviewer #2: Major Comments

• The paper talks about different factors affecting hospital births but should explain more about how spatial variability influence these factors.

• The methods section is clear, but it needs to explain:

a. How spatial weights are chosen in geographically weighted regression.

b. Why Kriging interpolation was used over other available methods?

c. If a sensitivity analysis done to check the accuracy of the results?

• The paper uses good spatial analysis, but it should clarify:

a. Why an adaptive kernel was used for GWR.

b. How the model handles the issue of locations being too similar to each other.

c. The importance of clusters in the Sat Scan study.

• The discussion compares findings with other studies but should explain more about why Mozambique exhibits such spatial variations.

• The conclusion should include suggestions on improving hospital birth rates.

Minor Comments

Rephrase the following sentences for clarity

Line 56: "Utilizing services provided …. health of mothers" … for example … Utilizing medical institutions for childbirth is a crucial strategy to enhance maternal health.

Line 63: "The World Health Organization identifies … causes of these fatalities"

Line 78: "On a worldwide scale, around 830 … pregnancy and childbirth."

Line 110: "The study in question utilized … Mozambique Demographic and Health Survey."

Line 123: "Mozambique’s administrative structure … provincial status" … for example, Mozambique consists of ten provinces and Maputo City, which has provincial status.

Line 133: "This variable was categorized into two: ‘home delivery’, which includes births took place at the respondent’s own home or another home, and ‘institutional delivery’..." → replace “took place” with “that occurred”

Line 147: "Data cleaning and preparation were conducted using STATA version 17 and Microsoft Excel … replace “conducted' with “performed”

Line 162: "A Moran’s I value close to -1 … no particular pattern." → A Moran’s I value near -1 suggests a dispersed distribution of institutional deliveries, indicating no specific pattern.

Line 222: "Less than half (46.76%) of women had a big problem to access healthcare institutions due to distance" → Less than half (46.76%) of women faced significant difficulties accessing healthcare institutions due to distance.

Line 252: "In the Kriging interpolation, … southern parts of Mozambique." → Kriging interpolation identified a high predicted prevalence of institutional delivery in the northeastern, central, and southern regions of Mozambique.

Line 331: "Despite global efforts, low utilization … and cultural factors." → Despite global efforts, institutional delivery services remain underutilized in many low- and middle-income countries due to socio-demographic, economic, and cultural factors.

Line 411: "Accordingly, hot spot regions for institutional delivery were detected in Maputo city, southwest Inhambane, central Sofala, southern Gaza, and western Niassa region." → Hotspot regions for institutional delivery were identified in Maputo City, southwest Inhambane, central Sofala, southern Gaza, and western Niassa.

Line 415: "Spatial regression analysis … emerging regions of Mozambique." → Spatial regression analysis identified wealth index, unwanted pregnancy, vehicle transportation, and the presence of skilled birth attendants as significant determinants of institutional delivery in Mozambique.

The study is methodologically sound and makes an important contribution to understanding institutional delivery in Mozambique. Clarifications in methodology, improved discussion structure, and a stronger conclusion would all considerably increase the manuscript's quality.

**Do you want your identity to be public for this peer review?** For information about this choice, including consent withdrawal, please see our Privacy Policy

Reviewer #1: No

Reviewer #2: No

---

## [Decision Letter · Decision Letter 1]

7 Aug 2025

Institutional delivery and its determinants among reproductive-age women in Mozambique: A Geographic weighted regression

PGPH-D-24-03063R1

Dear Mr. Zegeye,

We are pleased to inform you that your manuscript 'Institutional delivery and its determinants among reproductive-age women in Mozambique: A Geographic weighted regression' has been provisionally accepted for publication in PLOS Global Public Health.

Best regards,

Julia Robinson

Executive Editor

Reviewer Comments (if any, and for reference):

Reviewer's Responses to Questions

**Comments to the Author**

Reviewer #1: All comments have been addressed

publication criteria?

Reviewer #1: Yes

3. Has the statistical analysis been performed appropriately and rigorously?

Reviewer #1: Yes

4. Have the authors made all data underlying the findings in their manuscript fully available (please refer to the Data Availability Statement at the start of the manuscript PDF file)?

Reviewer #1: Yes

5. Is the manuscript presented in an intelligible fashion and written in standard English?

Reviewer #1: Yes

Reviewer #1: All the modifications suggested have been satisfactorily incorporated in the revised manuscript.

**Do you want your identity to be public for this peer review?** For information about this choice, including consent withdrawal, please see our Privacy Policy

Reviewer #1: No
